# Insomnia and depression levels among Malaysian undergraduate students in the Faculty of Medicine and Health Sciences (FMHS), Universiti Putra Malaysia (UPM) during Movement Control Order (MCO)

**Raja Muhammad Iqbal**[ID]ᵒ, **Nur Ilyana Binti Riza Effendi**[ID]ᵒ, **Sharifah Sakinah Syed Alwi**[ID]*ᵒ, **Hasni Idayu Saidi**[ID]ᵒ, **Seri Narti Edayu Sarchio**ᵒ

Faculty of Medicine and Health Sciences, Department of Biomedical Sciences, Universiti Putra Malaysia, Selangor, Malaysia

ᵒ These authors contributed equally to this work.

* sh_sakinah@upm.edu.my

**Data Availability Statement:** Data cannot be shared publicly because of concern of private

## Abstract

Rapid outbreak of coronavirus disease 2019 has caused the implementation of the movement control order (MCO) which aimed to reduce the spread in Covid-19 infections. While some may find it easy to adjust to the new norm, others found it difficult to switch from their normal routines and habits as according to the MCO SOP. This resulted in a more frequent insomnia and depression that subsequently impacted their mental health. Insomnia and depression levels are examined in this study as they relate to the Covid-19 Pandemic and the MCO among Malaysian undergraduate health sciences students at the Faculty of Medicine and Health Sciences, UPM. Random sampling methods were utilised with consideration of inclusion and exclusion criteria. The Patient Health Questionnaire-9 (PHQ-9) and the Insomnia Severity Index (ISI) were the instrument packages used in this investigation. An internet platform was used to distribute the questionnaire. Based on the results, it is concluded that depression and insomnia are significantly correlated, with a p-value of 0.05. This study also revealed the link between the severity of insomnia and the severity of depression among UPM students studying health sciences. The percentage of students with depression and insomnia was rather high (54.9% and 33.9%, respectively), and this occurred during the second wave of Covid-19 cases in Malaysia.

## Introduction

The World Health Organization named Covid-19 a global infectious disease in March 2020 [1]. To prevent the spread of this infectious disease, MCO was implemented under the 1967 Police Act and the Prevention and Control of Infectious Disease Act. The MCO is simply an order that prohibits mass gathering, travelling oversea and urges people to stay at home [2]. Although the objective of MCO is to flatten the curve on the Covid-19 cases, it still has negative

information of participants mental health status are to kept private other than the use for this research. Data are available from the Ethics Committee (contact via jkeupm@upm.edu.my) for researchers who meet the criteria for access to confidential data. The data underlying the results presented in the study are available from corresponding researcher Sharifah Sakinah Syed Alwi at sh_sakinah@upm.edu.my.

**Funding:** The research is funded by Universiti Putra Malaysia.

impacts on the psychological states and caused pressures not only due to the limitations and constraints in movements but also due to changing in their old routines and lifestyle as to adapt to the 'new norm' [3]. In certain worst-case scenarios, this could lead to mental health difficulties. This condition may cause discomfort and uneasiness due to rapid changes in the environment especially when the period of MCO/lockdown spans over 12 months.

In terms of population in Malaysia, 32.6 million as in 2021 with 91.8% being citizens and 8.2% non-citizens. The majority ethnic group among citizens was Bumiputera (67.4%), followed by Chinese (24.6%), Indians (7.3%) and others (0.7%). Malays were the predominant ethnic group in Peninsular Malaysia, accounting for 63.1% of citizens. In Sarawak, Iban made up 30.3% of citizens while in Sabah, Kadazan/Dusun made up 24.5% [4].

The Covid-19 pandemic hit Malaysia in January 2020, originating from three Chinese tourists who had entered Johor, Malaysia through Singapore. The first case began when the individual suspected to be infected with this virus while attending a conference in Singapore and met with several international delegates from China [5,6]. The second wave of Covid-19 in Malaysia began in February 2020, with a large cluster originating from a religious gathering [7]. The Movement Control Order (MCO) was implemented in March 2020 and successfully reduced the number of cases. However, the third wave of Covid-19 began in September 2020, due to the D614G-type mutation [8]. The total number of cases reached 169,379 by January 2021, with daily cases ranging from 6,000 to over 12,000 by June 2021. The health system was strained and a state of emergency was declared in January 2021, with a targeted MCO implemented. However, the new Covid-19 cases seem to increase significantly throughout the first half and third quarter of the year 2021 which forces the government to enforce Enhance Movement Control order to the localities that were severely affected with Covid-19 cases [9].

A person's mental state and ability to function in their environment is usually determined by the biological, psychological, and social factors [10]. Generally, mental health problems including depression and anxiety have been reported to have major a negative impact on the society during the Covid-19 pandemic [11]. For college students in particular, isolation enforcement in colleges, adjusting to virtual learning, social isolation, and uncertainty of the future may cause negative implications on their physical and mental health status [12].

The Covid-19 pandemic also had a psychological negative impact on Malaysian university students, with 20.4%, 6.6%, and 2.8% of them reporting mild to moderate, severe to extremely severe, and most severe anxiety levels, respectively [13]. According to Hetolang and Amone-P'Olak [14], depression in young adults is associated with poor academic performance, suicidal behaviour, and poor physical and mental health condition. Depression can negatively influence a person's feelings, thoughts, and behaviour [15]. Additionally, if depressive symptoms continue, this could result in major depressive disorder [16].

In moderate to severe cases of Major Depressive Disorder (MDD), changes in eating, sleeping, or weight are common symptoms, along with weariness, a decline in libido, difficulty concentrating, a sense of worthlessness, and recurrent thoughts of suicide [17]. It has also suggested a correlation between depression, anxiety, stress, family's socioeconomic and financial status [18]. Individuals with stable socioeconomic standing can satisfy the needs, which lessens the vulnerability to psychological and emotional discomfort [19]. Contrarily, students from lower socioeconomic backgrounds may be more susceptible to the conditions that lead to depression. Depression can also significantly impact sleeping patterns which lead to insomnia [20].

One of the key factors that contribute to efficient cognitive and emotional processing has been sleep. Gender and age are a risk factor for insomnia thus with older adults and women being more likely to experience insomnia. The cause of the higher risk in the elderly is unclear, but it may be connected to a population-specific partial loss of the sleep regulation system's functionality, which may result in insomnia [21].

Importantly, having multiple medical issues significantly increases the chance that an aged person would experience insomnia [22]. The 3-P approach (predisposing, precipitating, and perpetuating) is a good way to assess insomnia [23,24]. A lifelong propensity for stress-related sleep deprivation and a family history of insomnia are predisposing variables that raise the risk of acquiring insomnia. Medical, environmental, or emotional pressures can cause a pattern of insufficient sleep [25].

According to Marino et al. [26] there is a substantial correlation between sleeping habits and depression. Sleeping medications are often prescribed to depressed patients because insomnia and hyper-insomnia are thought to be significant indications of depression. Insomnia has been acknowledged as a severe public health problem that can cause depression to start as well as long standing mental and physical exhaustion, along with changes in temper, attentiveness, and memory [27]. According to meta-analysts by Baglioni et al. [27] and findings from Manber et al. [28] research, depression is two times more likely to develop in those who don't get enough sleep. Students that display these symptoms will therefore surely be at a disadvantage because it will have an impact on both their academic performance and future.

Additionally, anxiety and insomnia also can cause depression [27,29]. According to Akram et al. [30], anxiety symptoms can make it harder to fall asleep by increasing the rumination (which may be related to the stress of academic life for students). It could cause cognitive activity, agitation, and distress that are adversely toned, delaying the onset of sleep [31].

Based on Steger & Kashdan [32], rumination and overthinking are among the typical depressive symptoms. Rumination is the act of giving oneself or one's surroundings or the entire world a lot of thought. Beck postulated that depressed individuals had a propensity for negative evaluation. Beck defined the cognitive triad, negative self-schema, and logical fallacies as the three mechanisms he thought were responsible for depression [33]. We can identify the sort of rumination whether it goes toward severe depressive thoughts or just routine ruminating behaviour by using Beck's theory [34]. This may also be a sign of despair and anxiety, and it may also draw connections between sleeping habits and mental health because, according to a study by Dinis & Bragança, [35] unhappy students frequently ruminate on their problems late into the night.

Moreover, when there are significant changes in the environment and circumstance, routine alterations and subsequent behavioural changes become more apparent [36]. This is because routine has been occupying the mind and giving one's daily existence a sense of purpose [37]. Thus, altering one's habits or the environment may lead to negative effects on one's psychological state in which can be seen during the implementation of MCO. The MCO is a unique circumstance due to rapid significant changes and the longer duration of more than 12 months able to give impact to the student's behaviour.

Therefore, this study is important to examine and understand the relationship and implications between insomnia and depression levels among Malaysian undergraduate health science students at the Faculty of Medicine and Health Sciences (FMHS), UPM, during the Covid-19 Pandemic and MCO.
'

## Materials and methods

### Study population

The target population of this study were the undergraduate students from the health sciences programs composed of Doctor of Medicine, Biomedical Sciences, Environmental and Occupational Health, Science Dietetic, Science Nutrition and Community Health, and Nursing in the FMHS in Universiti Putra Malaysia throughout all their academic year of study regardless of

their as well as the socio-demographics factors. The survey is performed via an online via online platform. The total number of registered bachelor's degree students from health sciences programs during the time of survey distribution was 1,385. Sociodemographic factors such as gender, age, race, residential location, household accommodator, current residence, income stability, bachelor's degree, year of study, virtual learning, and region are independent variables for socio-demographic factors that can have implications toward depression and insomnia during the Covid-19 pandemic and MCO. This research is concentrating focusing on how socio-demographic characteristics during the Covid-19 pandemic and MCO might affect students' sleeping discrepancy (insomnia) and their degrees of depression.

## Research design

This quantitative cross-sectional study examined the relationship between undergraduate health sciences students' levels of depression and insomnia during the MCO. The instrument packages used in this investigation were the Patient Health Questionnaire-9 (PHQ-9) and the Insomnia Severity Index (ISI).

## Data source

The FMHS UPM in Malaysia was the site of the investigation during MCO and the data collection was completed within three months (between April and June 2021). The inclusion criteria were the undergraduate students from the FMHS, UPM that include Doctor of Medicine, Biomedical Sciences, Environmental and Occupational Health, Science Dietetic, Science Nutrition and Community Health, and Nursing. In addition, the exclusion criteria included foundation/matriculation, International, diploma, master & PhD students. The questionnaire was converted into an online format using Google form. A Respondent Information Sheet was added to the questionnaire's first page to explain the study's objectives and guarantee the participants' privacy and confidentiality. Students with PHQ-9 and ISI readings that fall into clinically severe and moderate categories or with any of the conditions respectively, will be informed via email and recommended to seek appropriate medical attention. In this study, 512 surveys in total were distributed, and of those, 472 were deemed to be usable, representing a 92% overall usable response rate.

## Sampling technique

Cross sectional sample size calculation was used in this study. To ensure that the intended group contained the appropriate percentage of students experiencing mental health issues, the sample size was estimated. The necessary minimum sample size (n = 195) was determined using the results from Islam et al. [12]. Due to sampling error in simple random sampling, n = 195 in this study was adjusted to the design effect by x2.5 [38], and the sample size was increased by 5% to strengthen the analysis, yielding n = 512. On top of the 195 required minimum respondents, a total of 512 questionnaires were sent out. All FMHS students' email addresses were gathered for the survey, which was done using a simple random sample technique. Using a random number generator, respondents were selected at random from the FMHS student email pool [39].

## Research tools/instruments

In this study, three components were used to create the questionnaire and it consisted of socio-demographic information, the PHQ-9 and ISI that were utilized as research tools.

The structure questionnaires asked in Section A were regarding the respondent's gender, age, race, nationality, residential location, household accommodator, current accommodation, income stability, bachelor's degree, year of studies, virtual learning, and region.

Section B [40] measured and diagnosed the severity of depression and other mental disorders that commonly occurred in primary care via PHQ-9 The Cronbach's Alpha testing consistency for PHQ-9 is 0.854, moreover the inter-rater scale was 0.87 indicates that the raters are scoring the individual's symptoms in a consistent and reliable manner, which is important for the validity and accuracy of the results [41]. The PHQ-9 contains nine questions that use a 4-point Likert scale. The range of the scoring system is 0–3. It consists of zero days, a few days, more than half of the days, and almost every day, each of which is worth 0 points, 1 point, 2 points, and 3 points, respectively. The total score ranges from 0–27. Total score 0–4 denotes no or very little depression, 5–9 mild depression, 10–14 moderate depression, 15–19 moderately severe depression, and 20–27 severe depression. The aggregate rating of the respondents provides a general indication of their mental health. Lower scores will be indicated by higher ratings for mental health.

ISI was included in Section C [3] It is a test to assess the type, extent, and impact of insomnia. Cronbach's Alpha test consistency is 0.90 [42]. The ISI has seven items that rate the severity of sleep maintenance, sleep onset, wakening issues, sleep dissatisfaction, interference with daytime performance from sleep disorders, and distress brought on by lack of sleep on a 5-point Likert scale. The range of the scoring system is 0 to 4. There are five categories: none, mild, moderate, severe, and very severe from 0 to 4 respectively. The total score ranges from 0–28. The aggregate score in some ways reveals the respondents' insomnia. The severity of the insomnia was indicated by higher ratings. The questionnaire was made available online and took roughly 15 minutes to complete.

## Ethical approval

The Ethic Committee for Research Involving Human Subjects (JKEUPM) of University Putra Malaysia in Malaysia officially approved this study with reference number JKEUPM-2020-198.

## Participant consent

Before the survey began, the students were given the option to participate voluntarily and their informed consent was obtained in written form. The confidentiality of their responses was guaranteed to remain confidential. The age range from the participant was 18 years old and above, the research doesn't involve minors as a participant.

## Statistical analysis

Software from IBM SPSS version 22 was used to examine the data (IBM Corporation, Armonk, NY, USA). The sociodemographic characteristics connected to depression and insomnia severity levels were identified using chi-square and ordinal regression techniques as shown in Fig 1. In the Ordinal Logistic Regression Analysis, all the variables that met the criteria for significance at the P-value 0.05 level in the chi-square tests were examined. Additionally, the association between the severity of insomnia and depression was examined using chi-square analysis.

## Results

Table 1 displays the respondents' demographic characteristics. 81.6% of the 472 respondents were females, 64.6% were between the ages of 21 and 23, and more than 35% were Bumiputra.

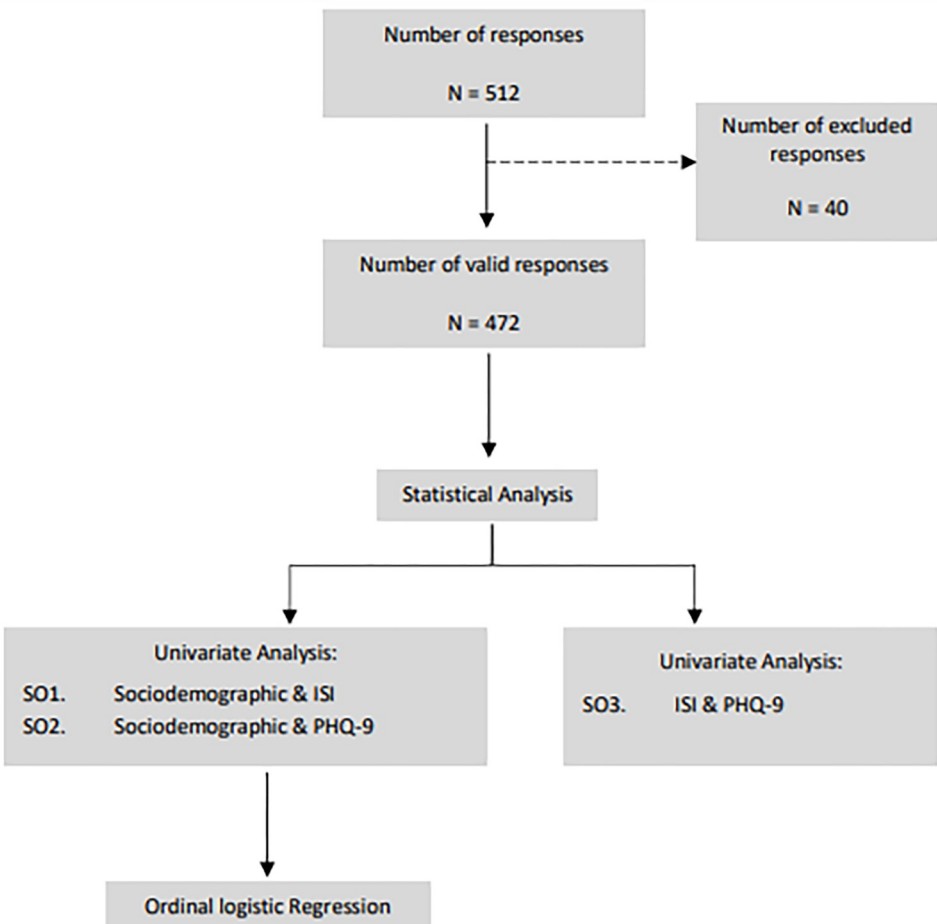

**Fig 1. Flow chart.** The chart shows the study's statistical analysis and data collection process. Specific objectives 1, 2, and 3 are denoted by SO1., SO2., and SO3, respectively.

Most of the students live in an urban region (79.9%), have a stable source of income (87.3%), and are living with their family.

Meanwhile, approximately 49.2% of participants were from the central region and 19.5% from the northern region, this makes up the majority of participants coming from Peninsular Malaysia. The region of Borneo was home to 4.4% of the participants in the research who were students. The majority of students 26.3% majored in biomedical sciences, followed by Doctor of Medicine (25%), environmental science and occupational health (18%), science nutrition and community health (11.7%), nursing (9.5%), and science dietetics (9.5%).

A majority 28.6% of responders were in their fourth year or older, followed by 24.2% in their first year, 24.8% in their second year, and 22.5% in their third year. At the time of data collection, 88.6% of students were taking classes online.

The 7-item Insomnia Severity Index (ISI) is thought to have an acceptable level of internal consistency (Cronbach's alpha = 0.882). According to the ISI, 164 (34.7%) of the sample's 472 respondents did not display any clinically significant insomnia, 148 (31.4%) showed sub-threshold insomnia, and 169 (35.8%) had moderately severe to severe insomnia (Table 2). Given low frequency in cases of "moderate severe insomnia" and "severe insomnia", both of the group were combined and given the new designation "moderate severe to severe" degree of insomnia.

**Table 1. Sociodemographic characteristic of the responses.**

| Variable | Frequency | Percentage (%) |
|---|---|---|
| **Gender** | | |
| Male | 87 | 18.4 |
| Female | 385 | 81.6 |
| **Age** | | |
| 18–20 Years | 97 | 20.6 |
| 21–23 Years | 305 | 64.6 |
| 24 Years and Above | 70 | 14.8 |
| **Race** | | |
| Bumiputra | 366 | 77.5 |
| Chinese | 65 | 13.8 |
| Indian | 41 | 8.7 |
| **Current area of residential** | | |
| Urban | 377 | 79.9 |
| Rural Area | 95 | 20.1 |
| **Household accommodator** | | |
| Alone | 48 | 10.2 |
| Family | 292 | 61.9 |
| Friends | 132 | 28.0 |
| **Current accommodation** | | |
| Outside campus | 344 | 72.9 |
| College (in campus) | 128 | 27.1 |
| **Financial stability** | | |
| Stable | 412 | 87.3 |
| Not stable | 60 | 12.7 |
| **Bachelor's Degree** | | |
| Biomedical Sciences | 124 | 26.3 |
| Doctor of Medicine | 118 | 25.0 |
| Environmental Science and Occupational Health | 85 | 18.0 |
| Nursing | 45 | 9.5 |
| Science Dietetic | 45 | 9.5 |
| Science Nutrition and Community Health | 55 | 11.7 |
| **Year of study** | | |
| 1st Year | 114 | 24.2 |
| 2nd Year | 117 | 24.8 |
| 3rd Year | 106 | 22.5 |
| 4th year and above | 135 | 28.6 |
| **Online virtual learning** | | |
| Yes | 418 | 88.6 |
| No | 54 | 11.4 |
| **Region** | | |
| Northern Region | 92 | 19.5 |
| Central Region | 232 | 49.2 |
| East Coast Region | 48 | 10.2 |
| Southern Region | 79 | 16.7 |
| Borneo Region | 21 | 4.4 |

**Table 2. Insomnia severity index and PHQ-9.**

|  | Frequency | Percentage (%) |
|---|---|---|
| **Severity of Insomnia** | | |
| No clinically significant insomnia | 164 | 34.7 |
| Subthreshold insomnia | 148 | 31.4 |
| Moderate severe to severe | 160 | 33.9* |
| **Severity of Depression** | | |
| None | 81 | 17.2 |
| Mild | 132 | 28.0 |
| Moderate to severe | 259 | 54.9** |

*Comprises both marked to moderate severe insomnia: 115 (24.4%) and severe insomnia: 45(9.5%).

**Comprises both marked to moderate depression: 135 (28.6%), moderate severe: 74(15.7%) and severe depression: 50(10.6%).

Additionally, the Patient Health Questionnaire-9's (PHQ-9) internal consistency was acceptable (Cronbach's alpha = 0.881). Based on the data from the PHQ-9, 81 (17.2%) of the 472 respondents showed no signs of depression, 132 (28%) were observed to have mild symptoms, and 259 (54.9%) had moderate to severe depression. In additional examination, "moderate," "moderate severe," and "severe depression" instances were combined due to low frequencies and given the new designation "moderate to severe" level of depression.

The data from the chi-square tests that demonstrated the correlation between sociodemographic factors and the severity of insomnia among the students in the faculty were shown in Table 3. The variables were gender, age, race, current residential region, current area of residential, financial stability, bachelor's degree, virtual learning and residential region. Significant sociodemographic indicators for the severity of insomnia included race (p = 0.002), household accommodator (p = 0.001), bachelor's degree (p = 0.024), and virtual learning (p = 0.001).

Table 3 demonstrated the Chi-square analysis (P-value 0.05) for ISI resulted in the selection of race, household accommodator, bachelor's degree, and virtual learning for the ordinal regression analysis (Table 4). The model fit was acceptable for the analysis (Deviance chi-square = 101.481, df = 84, p = 0.094). The equal-proportion assumption was met by the parallel lines test, which has a p-value of 0.083. According to Table 4, in comparison to Bumiputra students, Chinese students have 0.417 times (95% CI = (0.247, 0.704), Wald $\chi^2$ (1) = 10.712, p = 0.001) the odds of having moderate to severe insomnia as opposed to subthreshold or no insomnia. In addition, compared to students who lived with their families, students who lived alone have approximately 2.043 times (95% CI = (1.129, 3.7), Wald $\chi^2$ (1) = 5.568, p = 0.018) the odds of having moderate to severe insomnia as opposed to subthreshold or no insomnia.

Table 5 presented data from the chi-square analysis of the relationship between sociodemographic factors and the severity of depression among students studying health sciences in UPM. The variables were gender, age, race, current area of residential, household accommodator, current accommodation, financial stability, bachelor's degree, status of online study, and region of residential. A p-value of 0.05 or lower was regarded as statistically significant. The following factors were significant with severity of depression: gender (p = 0.01), race (p = 0.027), and home host (p = 0.023).

Gender, race and household accommodators were tested in the ordinal logistic regression analysis based on Chi-square analysis in Table 3 (P-value < 0.05) for PHQ-9. In the analysis, the model fit was acceptable (Deviance chi-square = 39.987, df = 29, p = 0.084). The p-value in the test of parallel lines was 0.407 which equal-proportion assumptions were met. As shown in

**Table 3. Chi-square analysis of sociodemographic and severity of insomnia.**

| Variable | No clinically Significant Insomnia | Subthreshold insomnia | Moderate to severe | Chi-square | p-value |
|---|---|---|---|---|---|
| **Gender** | | | | 1.229 | 0.541 |
| Female | 131(35.6%) | 125(32.2%) | 129(29.9%) | | |
| Male | 33(37.9%) | 23(26.4%) | 31(35.6%) | | |
| **Age** | | | | 6.449 | 0.168 |
| 18–20 Years | 31(32.0%) | 35(36.1%) | 31(30.9%) | | |
| 21–23 Years | 113(37.0%) | 84(27.5%) | 108(35.4%) | | |
| Above 24 Years | 20(28.6%) | 29(41.4%) | 21(30.0%) | | |
| **Race*** | | | | 16.821 | 0.002* |
| Bumiputra | 116(31.7%) | 123(33.6%) | 127(34.7%) | | |
| Chinese | 36(55.4%) | 15(23.1%) | 14(21.5%) | | |
| Indian | 12(29.3%) | 10(24.4%) | 19(46.3%) | | |
| **Current area of residential** | | | | 2.135 | 0.344 |
| Urban | 137(36.3%) | 116(30.8%) | 124(32.9%) | | |
| Rural Area | 27(28.4%) | 32(33.7%) | 36(37.9%) | | |
| **Household accommodator*** | | | | 19.968 | 0.001* |
| Alone | 9(18.8%) | 16(33.3%) | 23(47.9%) | | |
| Family | 91(31.2%) | 102(34.9%) | 99(33.9%) | | |
| Friends | 64(48.5%) | 30(22.7%) | 38(28.8%) | | |
| **Current accommodation** | | | | 5.475 | 0.065 |
| Outside campus | 112(32.6%) | 118(34.3%) | 114(33.1%) | | |
| College (in campus) | 52(40.6%) | 30(23.4%) | 46(35.9%) | | |
| **Financial stability** | | | | 6.530 | 0.088 |
| Stable | 148(35.9%) | 131(31.8%) | 133(32.3%) | | |
| Not stable | 16(26.7%) | 17(28.3%) | 27(45.0%) | | |
| **Bachelor's Degree*** | | | | 20.638 | 0.024* |
| Biomedical Sciences | 39(31.5%) | 34(27.4%) | 51(41.1%) | | |
| Doctor of Medicine | 54(45.8%) | 34(28.8%) | 30(25.4%) | | |
| Environmental Science and Occupational Health | 22(25.9%) | 32(37.6%) | 31(36.5%) | | |
| Nursing | 17(37.8%) | 18(40.0%) | 10(22.2%) | | |
| Science Dietetic | 10(22.2%) | 17(37.8%) | 18(40.0%) | | |
| Science Nutrition and Community Health | 22(40.0%) | 13(23.6%) | 20(36.4%) | | |
| **Year of study** | | | | 12.157 | 0.059 |
| 1st year | 26(22.8%) | 41(36.0%) | 47(41.2%) | | |
| 2nd year | 39(33.3%) | 40(34.2%) | 38(32.5%) | | |
| 3rd year | 45(42.5%) | 45(27.4%) | 32(30.2%) | | |
| 4th year and above | 54(40.0%) | 38(28.1%) | 43(31.9%) | | |
| **Online virtual learning*** | | | | 13.927 | 0.001* |
| Yes | 133(31.8%) | 136(32.5%) | 149(35.6%) | | |
| No | 31(57.4%) | 12(22.2%) | 11(20.4%) | | |
| **Region** | | | | 10.689 | 0.220 |
| Northern Region | 33(35.9%) | 24(26.1%) | 35(38%) | | |
| Central Region | 72(31%) | 74(31,9%) | 86(37.1%) | | |
| East Coast Region | 14(29.2%) | 18(37.5%) | 16(33.3%) | | |
| Southern Region | 35(44.3%) | 25(31.6%) | 19(24.1%) | | |
| Borneo Region | 10(47.6%) | 7(33.4%) | 4(19%) | | |

*Significant at 0.05.

**Table 4. Ordinal logistic regression analysis of sociodemographic factors and severity of insomnia (ISI).**

| Variable | B | Std.Error | Wald | df | Sig | 95% Confidence Interval | | OR | Odds 95% Confidence Interval | |
| --- | --- | --- | --- | --- | --- | --- | --- | --- | --- | --- |
| | | | | | | Lower Bound | Upper Bound | | Lower | Upper |
| **Race** | | | | | | | | | | |
| Indian | 0.469 | 0.318 | 2.166 | 1 | 0.141 | -0.155 | 1.093 | 1.598 | 0.856 | 2.982 |
| Chinese* | -0.874 | 0.267 | 10.712 | 1 | 0.001* | -1.398 | -0.351 | 0.417 | 0.247 | 0.704 |
| Bumiputra | *ref* | | | *ref* | | | | 1 | | |
| **Household accommodator** | | | | | | | | | | |
| Alone* | 0.715 | 0.303 | 5.568 | 1 | 0.018* | 0.121 | 1.308 | 2.043 | 1.129 | 3.7 |
| Friends | -0.385 | 0.21 | 3.362 | 1 | 0.067 | -0.797 | 0.027 | 0.68 | 0.451 | 1.027 |
| Family | *ref* | | | *ref* | | | | 1 | | |
| **Bachelor's Degree** | | | | | | | | | | |
| Biomedical Sciences | *0.432* | 0.303 | 2.030 | 1 | 0.154 | -0.162 | 1.025 | 1.540 | 0.850 | 2.788 |
| Doctor of Medicine | *-0.021* | 0.345 | 0.004 | 1 | 0.952 | -0.697 | 0.655 | 0.979 | 0.498 | 1.925 |
| Environmental Science and Occupational Health | *0.338* | 0.322 | 1.103 | 1 | 0.294 | -.293 | 0.970 | 1.403 | 0.746 | 2.638 |
| Nursing | *-0.062* | 0.383 | 0.026 | 1 | 0.871 | -0.813 | 0.689 | 0.940 | 0.444 | 1.991 |
| Science Dietetic | *0.474* | 0.376 | 1.590 | 1 | 0.207 | -0.263 | 1.211 | 1.607 | 0.769 | 3.356 |
| Science Nutrition and Community Health | *ref* | | | *ref* | | | | 1 | | |
| **Online virtual learning** | | | | | | | | | | |
| Yes | 0.446 | 0.370 | 1.452 | 1 | 0.228 | -0.279 | 1.171 | 1.562 | 0.756 | 3.227 |
| No | *ref* | | | *ref* | | | | 1 | | |

*Significant at 0.05.

Table 6, in comparison to male students, female students have 2.263 times (95% CI = 1.455, 3.519, Wald $\chi^2$ (1) = 13.141, p = 0.001) the odds of a moderate to severe versus mild or no depression. Meanwhile, in comparison to Bumiputra students, Indian students have 0.493 times (95% CI = 0.268, 0.905, Wald $\chi^2$ (1) = 0.023) the odds of a moderate to severe versus mild or no depression. Our further analysis also demonstrated that students who lived alone have 2.810 times (95% CI = 1.392, 5.672, Wald $\chi^2$ (1) = 0.004) odds of having moderate to severe depression as opposed to mild or no depression as compared to students who lived with their family.

Table 7 displays the results of the chi-square test that determined whether there was an association between the ISI and the PHQ-9 among health science university students in UPM. The severity level of both insomnia and depression was examined, and it ranged from none, mild, and moderate to severe depression to none, subthreshold sleeplessness, and moderate to severe insomnia. With a $p < 0.05$, this finding demonstrated a significant association between depression and insomnia.

## Discussion

The severe impacts of Covid-19 had affected numerous industries, whether they were in the commercial or public sectors. The loss of employment, income, routine alterations resulted in the need for adjustment to the new normal. A public health emergency has emerged as a consequence of the worldwide increase of Covid-19 infections and fatality rates. This had detrimental effects on people's psychological and mental health, including the university students. Since the beginning of the pandemic, universities and other educational institutions have been required to close as a precautionary step to further stop the spread of Covid-19 in educational

**Table 5. Chi-square analysis of sociodemographic and severity of depression.**

| Variable | None | Mild | Moderate to severe | Chi-square | p-value |
|---|---|---|---|---|---|
| **Gender*** | | | | 13.954 | 0.001* |
| Female | 57(14.8%) | 102(26.5%) | 226(58.7%) | | |
| Male | 24(27.6%) | 30(34.5%) | 33(37.9%) | | |
| **Age** | | | | 1.418 | 0.841 |
| 18–20 Years | 18(18.6%) | 25(25.8%) | 54(55.7%) | | |
| 21–23 Years | 54(17.7%) | 85(27.9%) | 166(54.4%) | | |
| Above 24 Years | 9(12.9%) | 22(31.4%) | 39(55.7%) | | |
| **Race*** | | | | 10.949 | 0.027* |
| Bumiputra | 52(14.2%) | 105(28.7%) | 209(57.1%) | | |
| Chinese | 17(26.2%) | 15(23.1%) | 33(50.8%) | | |
| Indian | 12(29.3%) | 12(29.3%) | 17(41.5%) | | |
| **Current area of residential** | | | | 3.409 | 0.402 |
| Urban | 70(18.6%) | 107(28.4%) | 200(53.1%) | | |
| Rural Area | 11(11.6%) | 25(26.3%) | 59(62.1%) | | |
| **Household accommodator*** | | | | 11.294 | 0.023* |
| Alone | 5(10.4%) | 6(12.5%) | 37(77.1%) | | |
| Family | 54(18.5%) | 84(28.8%) | 154(52.7%) | | |
| Friends | 22(16.7%) | 42(31.8%) | 68(51.5%) | | |
| **Current accommodation** | | | | 4.242 | 0.120 |
| Outside campus | 65(18.9%) | 89(25.9%) | 190(55.2%) | | |
| College (in campus) | 16(12.5%) | 44(33.6%) | 68(53.9%) | | |
| **Financial stability** | | | | 1.586 | 0.452 |
| Stable | 74(18.0%) | 113(27.4%) | 225(54.6%) | | |
| Not stable | 7(11.7%) | 19(31.7%) | 34(56.7%) | | |
| **Bachelor's Degree** | | | | 13.724 | 0.186 |
| Biomedical Sciences | 20(16.1%) | 30(24.2%) | 74(59.7%) | | |
| Doctor of Medicine | 29(24.6%) | 36(30.5%) | 53(44.9%) | | |
| Environmental Science and Occupational Health | 9(10.6%) | 23(27.1%) | 53(62.4%) | | |
| Nursing | 4(8.9%) | 14(31.1%) | 27(60.0%) | | |
| Science Dietetic | 7(15.6%) | 14(31.1%) | 24(53.3%) | | |
| Science Nutrition and Community Health | 12(21.8%) | 15(27.3%) | 28(50.9%) | | |
| **Year of study** | | | | 2.721 | 0.843 |
| 1st year | 17(14.9%) | 31(27.2%) | 66(57.9%) | | |
| 2nd year | 22(18.8%) | 34(29.1%) | 61(52.1%) | | |
| 3rd year | 20(18.9%) | 33(31.1%) | 53(50.0%) | | |
| 4th year and above | 22(16.3%) | 34(25.2%) | 79(58.5%) | | |
| **Online virtual learning** | | | | 0.690 | 0.708 |
| Yes | 70(16.7%) | 116(27.8%) | 232(55.5%) | | |
| No | 11(20.4%) | 16(29.6%) | 27(50.0%) | | |
| **Region** | | | | 2.665 | 0.954 |
| Northern Region | 17(18.5%) | 27(29.3%) | 48(52.2%) | | |
| Central Region | 39(16.8%) | 67(28.9%) | 126(54.3%) | | |
| East Coast Region | 6(12.5%) | 12(25%) | 30(62.5%) | | |
| Southern Region | 14(17.7%) | 20(25.3%) | 45(57%) | | |
| Borneo Region | 5(23.8%) | 6(28.6%) | 10(47.6%) | | |

* Significant at 0.05.

**Table 6. Ordinal logistic regression analysis of sociodemographic factors and severity of depression (PHQ-9).**

| Variable | B | Std.Error | Wald | df | Sig | 95% Confidence Interval | | OR | Odds 95% Confidence Interval | |
| --- | --- | --- | --- | --- | --- | --- | --- | --- | --- | --- |
| | | | | | | Lower Bound | Upper Bound | | Lower | Upper |
| **Gender** | | | | | | | | | | |
| Female* | 0.817 | 0.225 | 13.141 | 1 | 0.001* | 0.375 | 1.258 | 2.263 | 1.455 | 3.519 |
| Male | ref | | | ref | | | | 1 | | |
| **Race** | | | | | | | | | | |
| Indian* | -0.707 | 0.31 | 5.201 | 1 | 0.023* | -1.315 | -0.099 | 0.493 | 0.268 | 0.905 |
| Chinese | -0.489 | 0.259 | 3.578 | 1 | 0.059 | -0.997 | 0.018 | 0.613 | 0.369 | 1.018 |
| Bumiputra | ref | | | ref | | | | 1 | | |
| **Household accommodator** | | | | | | | | | | |
| Alone* | 1.033 | 0.358 | 8.307 | 1 | 0.004 | 0.331 | 1.736 | 2.810 | 1.392 | 5.672 |
| Friends | -0.013 | 0.203 | 0.004 | 1 | 0.949 | -0.412 | 0.386 | 0.987 | 0.662 | 1.47 |
| Family | ref | | | ref | | | | 1 | | |

*Significant at 0.05.

institutions. Even though the school was closed, the classrooms were transformed into virtual classes, with lecturers required to lead these classes virtually. With the updated MCO, SOP of the workplace and economic downturn, these changes create a sense of uncertainty about the future academic outlook and career prospects of the students [43].

This study was done as a web-based cross-sectional survey between April and June 2021, throughout the MCO period. Based on findings in Table 2, out of 472 respondents, 34.7%, 31.4%, and 33.9% of the students experienced subthreshold insomnia, moderately severe to severe insomnia, and no clinically significant insomnia, respectively. This data was quite alarming as more students were observed to have clinically significant insomnia than in the earlier studies done by Wang et al. [44], Marelli et al., [45]; and Yu et al., [46], which had respective prevalence rates of 27.2%, 16.3%, and 20% accordingly. However, a study done by Benham [47] in the United States showed a higher percentage of students with approximately 51.3% were reported to experience clinically serious insomnia. Moreover, the findings showed that the pandemic resulted in increased difficulty of falling asleep, increased use of sleep medication, and lower sleep efficiency [47]. Whereas, a study in Malaysia by Hasan & Moustafa [48] stated that the majority of students that were involved in their survey have poor sleep and that stress is a significant factor in the relationship between psychological distress and poor sleep quality. This may be due to the higher rate of sleeping problems experienced by the students during the Covid-19 pandemic, compared to the previous year [44–46]. Moreover, several scenarios were observed to be the factors that contributed to the increasing insomnia

**Table 7. Chi-square analysis of severity of Insomnia (ISI) and Severity of depression (PHQ-9).**

| | | PHQ-9 | | | Chi-square | P-value |
| --- | --- | --- | --- | --- | --- | --- |
| | | None | Mild | Moderate to severe | 144.482 | 0.0001* |
| **ISI** | No clinical significant Insomnia | 64 (39%) | 63 (38.4%) | 37(22.6%) | | |
| | Subthreshold insomnia | 13(8.8%) | 46(31.1%) | 89(60.1%) | | |
| | Moderate to severe | 4(2.5%) | 23(14.4%) | 133(83.1%) | | |

* Significant at 0.05.

among university students such as social isolation, living alone, anxiety of contracting Covid-19, and increased anxiety over pandemic scenarios [44,45].

Further analysis on the ordinal logistic regression analysis showed that only race and household accommodator were significant (p < 0.05). In comparison to the Chinese, Bumiputra students were 2.4 times more likely to experience insomnia. The findings also reported that Chinese students may have lower odds of developing insomnia than Malay students. This can be a reflection on the student's good sleep hygiene and sufficient sleep [49]. However, this finding contradicted a prior study conducted by Nurismadiana and Lee [50] in 2018, in the same institution and used a comparable sample of participants three years before the MCO struck. It was observed that there was no evidence of a significant relationship between students of different ethnic backgrounds and sleep quality reported. Furthermore, there are currently no definitive results from national and international research that indicated Bumiputra students in Malaysia are more likely to experience insomnia than other student ethnicities during the MCO Covid.

The most recent meta-analysis of sleep issues during the MCO was done by Alimoradi et al. [51] which revealed the limitations to generalising the findings to different ethnic groups which are due to the present review had a sizable proportion of Chinese and Italian populations because China and Italy are nations that had been severely affected by Covid-19.

Additionally, compared to students who lived with their family, students who live alone are 2.043 times more likely to experience insomnia. This finding is consistent with research done by Morin et al. [52], Morin et al. [53], and Bartoszek et al. [54], which found that people who live alone had a far higher chance of having insomnia than those who live with friends or family. These students are usually those who lived in a hostel or an apartment during the pandemic without friends or relatives who may be abroad or in another state. This situation may trigger elevated levels of vulnerability, loneliness, and weariness and therefore considered as one of the primary causes contributing to the worsening of insomnia. Similar conditions were also observed with the poor academic performance that triggered the stress that exacerbated their sleep problems [55,56]and worsened the insomnia [57]. There are also reports that relate loneliness as the contributing negative effect on sleeping patterns since it is closely related to psychological stress [57]. However, this study did not look at how stress affected loneliness.

It is common that staying alone means no social interaction with family, friends, and peers thus the possibility to develop depression and anxiety are higher [45]. The hypothalamic-pituitary-adrenal axis, which is linked to restless sleep, feelings of loneliness, and fear, may be the underlying molecular cause of this issue. A study by Gaultney [58] noted that up to 27% of children may be at risk for at least one sleep disorder and that adolescents with sleep issues are likely to perform poorly in school. Insomnia has been shown to be more frequent than usual in students who are at risk of failing their classes (GPA,2.0). Academic failure was a possibility for 30% of people who tested positive for obstructive sleep apnoea [59]. This is because a lack of sleep may result in an inability to concentrate in class. Obstructive sleep apnoea and hypersomnia are two of the most prevalent sleep disorders that impact both toddlers and adults.

The excessive use of screens late at night may also impact the quality of sleep, which is another worrying element that contributes to insomnia. This can be due to finishing late-night assignments and engaging in screen-based entertainment on a computer, tablet, phone, or other device. According to a higher Epworth Sleepiness Scale, less restful sleep, and fatigued driving have all been linked to computer use in the hour before bed [60]. Inability to fall asleep, frequent awakenings, and early morning awakenings have all been linked to using mobile devices before bed [61]. Moreover, developing good sleep habits may be difficult since they need to be disciplined in order to create a solid sleep routine and they also require social support from their peers and families in order to be motivated to get enough sleep [3].

In this study, it was also discovered that roughly 54.9% of the participants had severe depression. Since it includes more than half of the sample populations who experienced moderate to severe depression, this percentage is concerning. While 17.2% had no signs of depression and 28% displayed mild depression. Similar findings by Islam et al. [12], Hamaideh et al. [62] and Villani et al. [63] revealed that more than 50% of university students reported having depression during Covid-19 pandemic. These studies were conducted in Bangladesh, Jordan, and Italy, respectively. According to statistics, university students' rates of depression rose during the Covid-19 Pandemic compared to other years [12,62,63]. Additionally, the results of the ordinal logistic regression analysis (Table 7) revealed that the household accommodator, gender, and race were all significant with $p < 0.05$.

Meanwhile, the likelihood of developing depression among female students was 2.263 times more than male. These findings are consistent with data from previous research that reported women often have higher psychological symptoms related to the Covid-19 pandemic than men [64–66]. Similar report in Bangladesh was also observed from Islam et al. 2020 showing that female students are more likely to develop depression than male students. According to research, men are more resilient than women following stressful experiences, and women are more likely to develop stress and anxiety problems [67,68]. The present pandemic may have made this trend worse since women tend to show their emotions on average more than men. In addition, women have lower emotional tolerance levels than men, resulting in the development of stress and anxiety. Therefore, gender-specific mental health intervention strategies should be considered as a strategy to address differences in mental health between the sexes [69]. Additionally, Bumiputra students had the probabilities of being depressed by 2.028 times higher than Indian students. This data is consistent with studies conducted by Wong et al. [68] and Sundarasen et al. [13]. Although in our study, Malay and Bumiputra were grouped differently in which Bumiputra consisted of only Sabah and Sarawak, when we re-grouped the Malay with Bumiputra to match with grouping made by Wong et al. [68], the results showed similar significant association between severity of depression with distinct ethnicity.

Additionally, despite of the pandemic settings, this conclusion is concurrent with a previous study conducted seven years ago that used the same setting that was FMHS, Universiti Putra Malaysia (FMHS, UPM) [70]. Data obtained demonstrated strong association between the degree of depression among Bumiputra students and other ethnicities. Nevertheless, this study has limitations because of the small sample size obtained from other ethnic groups, such as Chinese and Indians, which caused noticeably higher rates of depression among Bumiputra during the Covid-19 epidemic. As reported earlier, it is generally known that there is a gender gap in the prevalence of depression around the world, with education, culture, and ethnicity having less of an impact than biological sex variations, which are thought to be the primary cause [71]. Moreover, students who lived alone had a 2.81 times higher chance of developing depression than those who did not. The results were consistent with that research done by Cooper et al. [72], Elmer et al. [73], and Wathelet et al. [74] which discovered a substantial difference and increased chance of depression severity while living alone during the Covid-19 pandemic. During the epidemic, lonely individuals who were separated from their families and friends may have lacked support, which made them feel even lonelier [54,75].

Regardless of the severity, prompt treatment is essential for treating depression since it will enable patients to recover from their depressive symptoms within the MCO period. Additionally, unfavourable ruminating may contribute to the students' depressed symptoms getting worse. Overthinking and ruminating have been identified as two frequent signs of depression by Steger and Kashdan [32]. Rumination is the act of giving oneself or one's relationships with others or the outside world a lot of thought. Beck hypothesised that depressed individuals tend to have negative evaluation on their experiences.

Theoretically, it was stated that depression is caused by three distinct mechanisms: the cognitive triad, negative self-schema, and logical fallacies. Based on Beck's premise, we may identify the type of rumination, such as whether it tends toward severe depressive thoughts or just regular rumination behaviour [34].

According to the results obtained (Table 7), there is a significant correlation between depression and insomnia ($p < 0.05$). This study reveals a possible correlation between ISI and PHQ-9. Our data are also in line with those studies carried out by Morin et al. [74], Raman, Hyland & Coogan [76] and Liu et al. [77]. One study found that approximately 83% of people with depression showed at least one sleep-related symptom, compared to 36% of those without depression [78]. If untreated, insomnia may potentially lead to depression. Delayed sleeping habits may also be a sign of depression. Insomnia can also result from extreme maladaptive behaviour brought on by long-term environmental changes, which can cause depression [79]. Approximately 97% of participants in research conducted by Nutt, Wilson, and Paterson [78] reported having sleep problems that began at the same time as their depression. Meanwhile, alteration in daily routine and environment may also lead to maladaptive behaviour [80]. Maladaptive to the new environment is a result of the inability to adjust and find purpose [37]. Additionally, according to a meta-analysis study done by Li et al. [81], there is a link between insomnia and a higher risk of depression. The results offer practical and helpful indications for depression prevention and aetiology study.

Rumination is also a cognitive predisposing factor for depression, according to Sun et al. [82], and it may aggravate the connection between depression and negative cognition. Additionally, they suggested that, among those with limited hope, excessive rumination is linked to severe depression and Insomnia can make negative thoughts more frequent, which can make depression symptoms worse [83]. Negative experiences can impair a person's mental health over time and contribute to or exacerbate depression. A person with high rumination repeatedly considers a situation in a negative way, magnifies the negative effect, grumbles frequently, and even decreases their self-worth; all these factors serve to exacerbate depression [84]. Therefore, as to prevent worsening situation and mental health, which could result in severe symptoms, mental decline, negative thoughts, self-harm, and suicide it is crucial to encourage student who have been struggling with both insomnia and depression to receive treatment and counselling as soon as they can. Nevertheless, it is important to clarify that the data collected and analysed in the study does not directly measure rumination among the population studied, however understanding rumination may provide better understanding on the predisposing factor for depression.

It is important to note that universities and other educational institutions have a significant impact on the intervention and play an important role to support the students who are suffering from mental health issues like depression, anxiety, stress, and insomnia. Digital and remote psychological assistance can be used to support affected individuals during MCO where the condition forbids face-to-face interaction with others. These tools include mental health awareness campaigns, counselling sessions and support groups via digital platforms. It is also crucial to ensure that issues raised by the current scenario are appropriately handled, changes or reviews to the curriculum and delivery method are required.

## Limitation

This study has some limitations that should be acknowledged. First, there is a drawback to the cross-sectional study design where associations are found but cause and effect cannot be determined because a follow-up survey has not been conducted. Second, the study demonstrates a lower percentage of Chinese and Indian students participating in the survey and a higher

percentage of female and Bumiputra participants in the sample. With more diverse participants, the sample disparity in terms of race and gender can be reduced while still providing a more accurate picture of the sample population of FMHS, UPM students. Furthermore, since the survey is done online and may be affected by the internet connection, user interface interpretation, and accessibility issues, bias in terms of computer literacy may need to be taken into account. The influence of Covid-19 on students using the IES-Covi19 (Event Scale with Modifications for Covid-19) was also not considered in this study because only the degree of depression and insomnia were assessed.

## Conclusion

The findings of this study imply the depression and insomnia that FMHS, UPM students were reportedly dealing with during the MCO. Even though there were few samples, the percentage of students who reported having depression or insomnia was rather high with approximately 54.9% and 33.9%, respectively, and this occurred during the implementation of MCO in Malaysia. This alarming situation may indicate that there will be a rise in mental health concerns among college students if they did not receive any proper treatment. The general deterioration of mental health status can contribute to depressive and insomnia symptoms. Despite the limitations mentioned previously, females and Bumiputra were more likely to experience depression whereas Bumiputra and living alone were more likely to experience insomnia. These findings highlight the significance of mental health status monitoring and psychological interventions that are particular and targeted to those who are experiencing mental health decline, as indicated in this study, with the aim of enhancing the students' mental health status throughout the Covid-19 pandemic period.

## Acknowledgments

We would like to acknowledge the participants, editors and reviewers.

## Author Contributions

**Conceptualization:** Nur Ilyana Binti Riza Effendi.

**Data curation:** Raja Muhammad Iqbal, Hasni Idayu Saidi.

**Formal analysis:** Raja Muhammad Iqbal, Hasni Idayu Saidi.

**Funding acquisition:** Sharifah Sakinah Syed Alwi.

**Methodology:** Raja Muhammad Iqbal, Nur Ilyana Binti Riza Effendi, Sharifah Sakinah Syed Alwi, Hasni Idayu Saidi.

**Software:** Raja Muhammad Iqbal.

**Supervision:** Sharifah Sakinah Syed Alwi.

**Validation:** Sharifah Sakinah Syed Alwi.

**Visualization:** Raja Muhammad Iqbal.

**Writing – original draft:** Raja Muhammad Iqbal.

**Writing – review & editing:** Sharifah Sakinah Syed Alwi, Hasni Idayu Saidi, Seri Narti Edayu Sarchio.

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
