## [Decision Letter · Decision Letter 0]

19 Dec 2022

PONE-D-22-29913Implication of Insomnia and Depression Among Malaysian Undergraduate Health Science University Students in UPM During Covid-19/Movement Control OrderPLOS ONE

Dear Dr. Bin Raja Yahya,

Thank you for submitting your manuscript to PLOS ONE. After careful consideration, we feel that it has merit but does not fully meet PLOS ONE’s publication criteria as it currently stands. Therefore, we invite you to submit a revised version of the manuscript that addresses the points raised during the review process.

 Please see reviewer's comments below and submit the edited manuscript accordingly. 

We look forward to receiving your revised manuscript.

Kind regards,

Ankit Jain, M.D.

Academic Editor

PLOS ONE

Journal Requirements:

4. PLOS requires an ORCID iD for the corresponding author in Editorial Manager on papers submitted after December 6th, 2016. Please ensure that you have an ORCID iD and that it is validated in Editorial Manager. To do this, go to ‘Update my Information’ (in the upper left-hand corner of the main menu), and click on the Fetch/Validate link next to the ORCID field. This will take you to the ORCID site and allow you to create a new iD or authenticate a pre-existing iD in Editorial Manager. Please see the following video for instructions on linking an ORCID iD to your Editorial Manager account: https://www.youtube.com/watch?v=_xcclfuvtxQ.

5. Please remove your figures from within your manuscript file, leaving only the individual TIFF/EPS image files, uploaded separately. These will be automatically included in the reviewers’ PDF.

6. Please include a copy of Table 8 which you refer to in your text on page 20.

Reviewers' comments:

Reviewer's Responses to Questions

**Comments to the Author**

1. Is the manuscript technically sound, and do the data support the conclusions?

Reviewer #1: Yes

Reviewer #2: Yes

Reviewer #3: Yes

Reviewer #4: Yes

2. Has the statistical analysis been performed appropriately and rigorously? 

Reviewer #1: Yes

Reviewer #2: Yes

Reviewer #3: I Don't Know

Reviewer #4: Yes

3. Have the authors made all data underlying the findings in their manuscript fully available?

Reviewer #1: Yes

Reviewer #2: No

Reviewer #3: Yes

Reviewer #4: Yes

4. Is the manuscript presented in an intelligible fashion and written in standard English?

Reviewer #1: Yes

Reviewer #2: Yes

Reviewer #3: Yes

Reviewer #4: Yes

5. Review Comments to the Author

Reviewer #1: Thank you for shedding light on this topic. This paper has a remarkable potential to bring this issue into the mainstream through PLOS, but I suggest some minor revisions.

The sample demographics are markedly skewed regarding gender (female>> male). The sample likely reflects the heterogeneous nature of the study population.

By design of its proposed testing methodology, PHQ-9 captures symptomatology over the past fourteen days, which varies significantly based on interrater reliability. Consider commenting on inter-rater reliability in the methodology section.

The introduction is well-framed and discusses Depression and Insomnia in terms of clinical presentation, known etiology and neurobiology.

Paper could benefit from adding any specific resources, if they exist (CDC, AACAP, AAP, APA, etc.), that clinicians can access to learn more on it or seek guidance.

Reviewer #2: This is a cross-sectional study performed to examine a relationship between undergraduate students' level of depression, insomnia during lockdown in Malaysia. This well-done study identifies several key features that can allow clinicians to identify risk factors among undergraduate students in order to diagnose and treat mental health issues. The study appropriately identifies the limitations it has. The conclusion section of the article can be further modified in order to provide concrete, actionable knowledge that can be acted upon by governments and universities to further help their students.

This study does not provide context in regard to the COVID-19 pandemic in Malaysia in the introduction section. While COVID-19 pandemic affected the entire globe, every country had their own individual experiences with COVID-19 spread. Some countries had an exceptionally severe first wave, while others had a terrible second wave. The authors can improve this article by providing context about how COVID-19 pandemic played out in Malaysia. The same can be said about race and different regions of Malaysia from which the students come from as western readers will be unsure what differentiates one region and race of Malaysia from the other socio-demographically. This will give readers further information about whether the undergraduate students were prepared to face the challenges they will face if COVID-19 spreads in their country.

Another concern observed is the lack of discussion of adjustment disorder with depressed mood and acute stress disorder as differential diagnoses for these students. COVID-19 pandemic can be looked at as a significant event, which can lead to depressive features in an individual who is unable to adjust to this significant change in their lives.

The second paragraph of discussion compares the higher rate of insomnia in students in the United States with the findings of the study. The authors can further discuss the difference in the US study to that in Malaysia in order to further explore the reasons behind this difference in students being affected.

The authors can also consider adding references to allow Western readers to understand the differences between races in Malaysia in order to appreciate the third paragraph of discussion.

The authors identify and discuss excessive use of screens late at night as an issue leading to insomnia. They can also consider discussing the COVID-19 related increased social media usage being a cause. The initial wave of the pandemic was associated with an "infodemic" of fear mongering and harmful misinformation propagated from social media. This led to a lot of people falling victim to quackery and unscientific methods. This was also politicized in many different countries all over the globe, and it gave rise to conspiracy theories and echo chambers where people magnified each other's anxiety and paranoia.

Reviewer #3: The study raises important questions about impact of COVID on the depression and insomnia, on the students. The does have some limitations as to it is unclear, how that the impact of Covid on the outcome measures is assessed. The conclusion talks about female gender, Bumiputra, and people living alone being more at risk of depression and insomnia, although it might be helpful to know how much different was the data, pre covid and post covid for these populations. It does raise awareness about importance of mental health support for college students, through social media, support groups and generating awareness through internet.

Reviewer #4: Hello. Thank you for submitting your manuscript for review. It was a good read and interesting to note the conclusions/ results you had. I had only certain minor suggestions:

1) Page 2 - "The most prevalent condition in terms of depression among teenagers and young adults

is Major Depressive Disorder (MDD)" - The most prevalent condition is MDD? I suspect that there might be less severe depressive symptoms that might be more prevalent.

2) Page 2-"One of the key factors that contribute to efficient cognitive and emotions process has

been sleep. Insomnia is a risk factor for both gender and age, with older adults and

women being more likely to experience insomnia." - Both of these sentences can be reworded to be more grammatically accurate. e.g. processing instead of process; Insomnia is a risk factor? or probably better called- older Age and female gender are risk factors for insomnia.

3) Page 3 - "In addition, anxiety and insomnia also can cause depression [21, 23]. According to Akram

et al. [24], anxiety symptoms can make it harder to fall asleep by increased the

rumination (which may be related to the stress of academic life for students)." Please change increased to increasing

4) In results (Chart in page 16) - I'm curious why were 'mild/ moderate insomnia' not listed and were considered 'subthreshold insomnia'.

5) In discussion (page 17)- I can see how you are comparing the rates you had with rates in studies done elsewhere but you can't really compare those studies because of various different confounding factors in those samples. Since you had a study from similar population from by Sundarasen S, et. al., that could have been a better comparison. Although I don't think you took any significant conclusion from this comparison as such.

6) Could you discuss why Bumiputra students in this study had more likelihood of having depression.

7) What was the purpose of discussing rumination as a separate entity when your data uses PHQ-9 and ISI and does not measure rumination among the population you studied?

Again,

Thank you for giving us the opportunity to review your work and putting so much effort into studying and writing it.

6. PLOS authors have the option to publish the peer review history of their article (what does this mean?). If published, this will include your full peer review and any attached files.

Reviewer #1: No

Reviewer #2: **Yes: **Lakshit Jain MD

Reviewer #3: **Yes: **Meenal Pathak

Reviewer #4: No

---

## [Author Response · Author response to Decision Letter 0]

21 Feb 2023

Dear Editor,

We thank the reviewers for their generous comments on the manuscript. On behalf of all the authors, I would like to submit rebuttal letter of “Implication of Insomnia and Depression Among Malaysian Undergraduate Health Science University Students in UPM During Covid-19/Movement Control Order” study in response to academic editor comments on this study revision.

This letter will address to journal requirements (Section A) and reviewers’ comment (Section B) as stated in “PLOS ONE Decision: Revision required [PONE-D-22-29913] - [EMID:46a201b3f656365c]” email, answered according to the question order. 

Section A 

1. The manuscript will comply to the PLOS ONE's style requirements.

2. Additional details of participant consent were added in the manuscript (Page 7, 2nd paragraph).

3. Data from this study are available upon request, details were noted in the cover letter. 

4. ORCID ID for primary author Raja Muhammad Iqbal Bin Raja Yahya ORCID ID: 0000-0002-0550-6751 and corresponding author Sharifah Sakinah Syed Alwi ORCID ID: 0000-0002-6497-706X

5. Figure were removed in the manuscript, figure 1 in page 4 were deleted (Conceptual framework) in replaced to “Population Study” level 2 heading. In note to the deletion that Figure 2 in page 8 under “Statistical analysis” level 2 heading will change to Figure 1.

6. Typing error, the origin intend was to cited Table 7 in page 21.

7. Reference list were reviewed and updated to relevant current references. However if there is an reference error please do contact us so amendment can be made.

Section B 

Reviewer #1: The sample demographic is the relative representation of the student population, though the population of female student is more compared to male student it is true reflection of gender disparity in public university in Malaysia where the numbers of female student outweighs the number of male student (Tienxhi, 2017)

Tienxhi, J. Y. (2017). The Gender Gap in Malaysian Public Universities: Examining the'Lost Boys. Journal of International and Comparative Education (JICE), 1-16.

Moreover, Inter-rater reliability was noted in the manuscript within the methodology section (Page 6, 5th paragraph). 

Reviewer #2: Amendment were made based on the comments. 

Reviewer #3: Differentiation level of depression and insomnia data, pre covid and post covid for these populations were not perform in this study as it only measure depression and insomnia level during the movement control order/covid-19 pandemic. 

Reviewer #4: Amendment were made based on the comments. 

For comment No.4 'mild/ moderate insomnia' were not used because the classification of insomnia are based from the classification from the origin paper from Morin et al., 2011.

Morin CM, Belleville G, Bélanger L, Ivers H. The insomnia severity index: Psychometric Indicators to detect insomnia cases and evaluate treatment response. Sleep. 2011;34(5):601–8. 

For comment No.5 the reason why, the rates were compared study done outside Malaysia because during the time of writing there are limited studies in Malaysia that addressing depression and insomnia in university students. Moreover, rates comparison to Sundarasen et al., 2020 were unable to be made because the paper focuses on the anxiety levels of student during the lockdown. 

For comment No.7 rumination was discussed as a separate entity for theoretical or conceptual reasons, or to provide background or context for the study.

In conclusion, I have presented clear evidence and arguments refuting the claims made by the reviewers in their evaluation of our paper. I believe that our research is scientifically sound and deserves to be published in PLOS ONE. I hope that after reviewing my rebuttal, the publisher will recognize the validity of our findings and make the necessary corrections to the paper.

Thank you for considering this study and I look forward to your positive response.

Sincerely,

Raja Muhammad Iqbal Bin Raja Yahya

---

## [Editor Report · Decision Letter 1]

2 Mar 2023

Implication of Insomnia and Depression Among Malaysian Undergraduate Health Science University Students in UPM During Covid-19/Movement Control Order

PONE-D-22-29913R1

Dear Dr. Bin Raja Yahya,

We’re pleased to inform you that your manuscript has been judged scientifically suitable for publication and will be formally accepted for publication once it meets all outstanding technical requirements.

Kind regards,

Ankit Jain, M.D.

Academic Editor

PLOS ONE

---

## [Editor Report · Acceptance letter]

2 Oct 2023

PONE-D-22-29913R1 

Insomnia and depression levels among Malaysian undergraduate students in the Faculty of Medicine and Health Sciences (FMHS), Universiti Putra Malaysia (UPM) during Movement Control Order (MCO) 

Dear Dr. Iqbal:

I'm pleased to inform you that your manuscript has been deemed suitable for publication in PLOS ONE. Congratulations! Your manuscript is now with our production department. 

Kind regards, 

on behalf of

Dr. Ankit Jain 

Academic Editor

PLOS ONE